# Matrices of Native and Oxidized Pectin and Ferrous Bisglycinate and Their *In Vitro* Behavior through Gastrointestinal Conditions

Martin Jimenez [1,2], Daniela Viteri [2], Daniela Oña [2], Marco Leon [3], Valeria Ochoa-Herrera [4,5,6], Natalia Carpintero [4], Francesc Sepulcre [1] and Jose F. Alvarez-Barreto [2,7,*]

1    Escola de Doctorat de Tecnologia Agroalimentària i Biotecnologia, Departament d'Enginyeria Agroalimentària i Biotecnologia, Universitat Politècnica de Catalunya, Campus del Baix Llobregat, 08860 Barcelona, Spain
2    Biomaterials Laboratory, Department of Chemical Engineering, Colegio de Ciencias e Ingenierias, Universidad San Francisco de Quito, Pichincha 170901, Ecuador
3    Department of Mechanical Engineering, Colegio de Ciencias e Ingenierias, Universidad San Francisco de Quito, Pichincha 170901, Ecuador
4    Core Laboratory—Environmental Engineering, Colegio de Ciencias e Ingenierias, Universidad San Francisco de Quito, Pichincha 170901, Ecuador
5    Escuela de Ingeniería, Ciencia y Tecnología, Universidad del Rosario, Bogotá 111221, Colombia
6    Environmental Sciences and Engineering, Gillings School of Global Public Health, University of North Carolina at Chapel Hill, Chapel Hill, NC 2759, USA
7    Institute of Energies and Materials, Diego de Robles y Pampite S/N, Hayek 104H, Campus Cumbayá, Universidad San Francisco de Quito, Pichincha 170901, Ecuador
*    Correspondence: jalvarezb@usfq.edu.ec; Tel.: +593-297-1700 (ext. 1328)

**Abstract:** Colloidal matrices of native and oxidized pectin were developed to improve iron bioavailability through the digestive tract. Ferrous bisglycinate (Gly-Fe), obtained by precipitation of glycine chelation to $Fe^{2+}$, was mixed with native and peroxide-oxidized citrus pectin, and subsequently lyophilized. Controls included matrices with iron and glycine without chelation. The resulting samples were characterized through FTIR, SEM, and TGA/DSC before and after in vitro digestion, which was performed in simulated salivary, gastric, and intestinal fluids. During these digestions, swelling capacity and iron release were assessed. All matrix formulations were porous, and while pectin oxidation did not alter architecture, it changed their properties, increasing thermal stability, likely due to greater number of interaction possibilities through carbonyl groups generated during oxidation. This also resulted in lower swelling capacity, with greater stability observed when using the chelated complex. Higher swelling was found in gastric and intestinal fluids. Pectin oxidation also increased retention of the chelated form, contrary to what was observed with unchelated iron. Thus, there is an important effect of pectin oxidation combined with iron in the form of ferrous biglyscinate on matrix stability and iron release through the digestive tract. These matrices could potentially improve iron bioavailability, diminishing organoleptic changes in fortified iron foods.

**Keywords:** citrus pectin; modified pectin; ferrous bisglycinate; iron transport; matrix digestion

## 1. Introduction

Iron bioavailability in the body is not completely guaranteed because components in food may entrap iron and other metal cations, resulting in health problems such as anemia. Phytic acid is not digestible for humans or animals, and it is known to be antinutritive as an iron adsorption inhibitor in soy-based foods such as soy milk, tempeh, and tofu, as well as in grains, legumes, oilseeds, and nuts [1]. Phytic acid and iron become insoluble complexes that are not available for adsorption under pH conditions in the small intestine, thereby contributing to a deficiency of iron bioavailability [2,3].

Highly bioavailable iron sources (ferrous sulfate and ferrous fumarate) are soluble in neutral or acidic aqueous environments but can cause organoleptic changes that result in low acceptability and shortened product shelf life [4]. Thus, this problem has been tackled through mineral chelation with an amino acid, most commonly glycine ($C_2H_5NO_2$), which has a structure strong enough for mineral protection in digestive process, and weak enough for mineral release in the intestine. Iron supplements produced with this method have been successful in reducing iron deficiency anemia in pregnant women [5]. Glycine's chelating capacity can also be used with various metallic cations such as zinc and copper, among others [6]. Most importantly, the high bioavailability of chelates in digestive absorption processes has been demonstrated in several cases, such as iron bisglycinate as an alternative in the prevention and treatment of anemia in newborns [7,8].

Ferrous bisglycinate is a source of iron chelated with glycine, but similarly to other water-soluble iron compounds, it is readily oxidized to $Fe^{3+}$, leading to off-color development and fat oxidation when added to foods and beverages [4,9,10]. A higher pH favors the formation of $Fe^{3+}$ over $Fe^{2+}$. Therefore, encapsulation-based technologies may be used to solve iron-mediated problems in products above pH 5. Thus, iron is rendered nonreactive during formulation and storage, isolating it from carrier components such as tannins, fats and vitamins, and oxidizing agents such as dissolved oxygen and chlorine [11].

An alternative method is the generation of colloidal forms obtained through ionic gelation of carbohydrates, such as pectin, with $Fe^{2+}$. Pectins with a low degree of esterification (<50%) are useful for ionotropic gelation. As this polysaccharide is a dietary fiber, it is not hydrolyzed in the upper gastrointestinal tract; consequently, colloidal pectin matrices have been widely used to deliver drugs that target the colon [12,13]. The maximum stability of pectin occurs at pH 4, which is when it begins to lose methoxyl groups and hydrolyze, losing its ability to form gels [14,15].

Ionotropic gelation has been mainly explored with calcium and zinc, and little has been done with the use of iron. However, studying ionotropic pectin gelation with iron is relevant, as most of the iron ingested is absorbed in the duodenum as well as in the colon, where the mucosa also possesses iron absorption proteins which allows for the absorption of 30% of the iron passing through the gastrointestinal tract. Indigestible carbohydrates, like pectin, resist digestion in the small intestine, but are fermented in the colon to form short-chain fatty acids, with some health benefits, including improved iron absorption [16].

Moreover, pectin can be used in its native form or modified, with the latter having different properties. Several types of products have been created based on pectin modification, depending on the degree of methoxylation [17], the addition of carbon chains to carboxyl groups [18], and oxidation, among others. Pectin oxidation creates carbonyl groups in the polymer's backbone, which could form imines with amino groups [19] through Schiff bases.

It is believed that colloidal structures formed through ferrous bisglycinate and oxidized pectin could be strengthened through the ionotropic gelation and Schiff bases between the amino group of glycine and carbonyl groups in oxidized pectin. Therefore, the present work aimed to study the combined effect of these two interactions on matrix structure, stability, and ability to retain iron through the digestive tract. Through the use of a matrix of previously oxidized pectin as a carrier for ferrous bisglycinate, it would be possible to find a mechanism to strengthen iron stability and avoid its oxidation in fortified foods; furthermore, it is expected that, by using this matrix, iron bioavailability through the digestive tract until the colon would be improved. The development of these matrices could result in a technology capable of transporting iron, in its amino–chelated form, within the body and achieve prolonged bioavailability and controlled release in specific areas of the digestive tract, such as the duodenum and colon [20].

## 2. Materials and Methods

The main materials used in the study were L-glycine (purity 99%), $FeSO_4.7H_2O$ as iron source, commercial food grade citrus pectin with a low methoxylation degree, 30% *v/v* hydrogen peroxide, and 96% *v/v* ethanol, provided by Core Laboratory-Environmental

Engineering. Colegio de Ciencias e Ingenierias, Universidad San Francisco de Quito USFQ, Quito, Ecuador.

### 2.1. Ferrous BisGlycinate Sample Chelation

First, ferrous bisglycinate samples were obtained by precipitation, according to the method proposed by Yunarti et al., 2013, with another type of molar relationship between glycine and iron of 1:1. Glycine was codissolved with 80% *w/v* $Fe_2SO_4$ in distilled water, and the pH was adjusted below 2.34, which is the pKa of the glycine carbonyl group [1].

Nitrogen was injected into hermetically sealed flasks for 5 min, under continuous agitation to favor the crystal formation [1]. Then, the flasks were kept under continuous agitation at 50 °C for 24 h, and later refrigerated at 4 °C until crystal precipitation, between 8 to 11 days, depending on precipitation level. Finally, crystals were washed with 96% ethanol, filtered, and dried at 40 °C.

### 2.2. Pectin Characterization

The degree of methoxylation refers to the number of esterified carboxyl groups present in polygalacturonic acid of pectin. To establish free and esterified carbonyl groups, a 1% *w/v* solution of pectin in water at 40 °C under constant stirring was titrated potentiometrically with 0.1M NaOH [14,15]. To quantify free carboxyl percentage, Equation (1) was used.

$$Kf = N(NaOH) \times V(NaOH) \times 0.045)/a \times 100 \qquad (1)$$

where, Kf is free carboxyl percentage, *a* is weight of sample, N (NaOH) is sodium hydroxide molar concentration, and V (NaOH) is volume used for titration (mL).

After the first titration, 10 mL of 0.1 M NaOH solution was added, and the mixture was stirred for 2 h at room temperature. Subsequently, excess base was neutralized with 10 mL of 0.1 M HCl and titrated with 0.1 M NaOH. Esterified carboxyl percentage was calculated through Equation (2).

$$Ke = N(NaOH) \times V(NaOH) \times 0.045)/a \times 100 \qquad (2)$$

where, Ke is esterified carboxyl percentage, *a* is samples weight, N(NaOH) is sodium hydroxide molar concentration, and V(NaOH) is volume used for the titration.

Total carboxyl groups was calculated as the sum of Kf (1) and Ke (2) as % *w/v*.

### 2.3. Pectin Oxidation

Pectin was oxidized to obtain free carbonyls (aldehydes) that can react with ferrous bisglycinate amino groups. Hydrogen peroxide ($H_2O_2$) was used as the oxidizing agent, according to method proposed by [21]. Briefly, pectin was dissolved in 2% $H_2O_2$ at a final pectin concentration of 2% *w/v*. The reaction was carried out for 30 min, at pH 1, under constant stirring and protected from light. Oxidized pectin was precipitated with 96% ethanol, for 24 h. The precipitate was filtered and washed with distilled water, and dried at 40 °C [19].

Aldehyde quantification was used as a parameter to determine oxidation efficiency, through the methodology reported by Siggia & Maxcy, 1947, which is based on a reaction with sodium sulfite in an acidic medium and a potentiometric titration with a strong base. Briefly, 0.2 g of sample was dissolved in 5 mL of 1M sulfuric acid and 25 mL of 1M sodium sulfite. The mixture was titrated potentiometrically to turn with NaOH [22]. Aldehyde concentration was determined using a standard curve generated with known solutions of butyrylaldehyde.

### 2.4. Preparation of Pectin, Glycine, Iron, and Ferrous Bisglycinate Matrices

The interactions between pectin (native and oxidized), glycine, and iron were studied in 3 ways: first, making matrices of native pectin and iron. Second, previously obtaining ferrous bisglycinate crystals by precipitation [1], to then interact in solution with pectin,

and third, making matrices of pectin, glycine, and iron, without prior chelation, and mixing them for a certain time. The resulting mixtures were then frozen and lyophilized, obtaining aerogel-like matrices; in this study single beads were produced.

To link oxidized pectin carbonyl groups with glycine amino groups, it was necessary to form a Schiff base at pH 5 [23]. Stability of Gly-Fe and iron oxidation were evaluated at pH 3.5, where half of pectin carboxyl groups are ionized; then, competition between link of iron to carboxyl and formation of Schiff base were compared. Tests were carried out according to formulations defined in Table 1.

**Table 1. Matrix compositions.** Pectin (Pect), oxidized pectin (Pectox), iron (Fe), Glycine (Gly), Ferrous Bisglycinate previously chelated (Gly-Fe).

|  | Pect + Fe | Pectox + Fe | Pect + (Gly-Fe) | Pectox + (Gly-Fe) | Pect + Fe + Gly | Pectox + Fe + Gly |
|---|---|---|---|---|---|---|
| **Concentration (*w/v* %)** | 10 + 10 | 10 + 10 | 10 + 10 | 10 + 10 | 10 + 5 + 5 | 10 + 5 + 5 |

There were 4 replicates per sample.

### 2.5. In Vitro Digestion

Lyophilized samples were digested in simulated gastrointestinal solutions in order to characterize the structure, stability, and iron retention until its intestinal digestion in the colon. Briefly, 50 mg of sample was suspended for 2 min in 1 mL saliva (10 mg/mL Bacillus subtilis $\alpha$-amylase) in phosphate buffered saline (PBS) (K2HPO4 0.144 g/L; NaCl 9.00 g/L; Na2HPO4 0.795 g/L, pH 6.8). Saliva was then removed, and 1 mL of simulated gastric digestion fluid (3 mg/mL porcine pepsin, 125 mM NaCl, 7 mM KCl, 45 mM NaHCO3, pH 2.5) was added to the samples. Samples were incubated for 1.5 h at 37 °C under shaking at 50 rpm. Subsequently, gastric juice was removed, and 1 mL of simulated intestinal fluid was added (1 mg/mL pancreatin, 1.5 mg/mL bile salts, 22 mM NaCl, 3.2 mM KCl, 7.6 mM NaHCO3, pH 8.0) to samples and incubated for 3 h at 37 °C, with continuous gentle shaking (50 rpm) [16]. Once each of the digested samples was obtained, these were frozen and lyophilized, and were then analyzed using Scanning Electron Microscopy (SEM), Fourier transform infrared spectroscopy (FTIR), thermogravimetric analysis (TGA), differential scanning calorimetry (DSC), for degree of swelling and iron release analysis.

### 2.6. Scanning Electron Microscopy (SEM)

Previously frozen and lyophilized samples were analyzed in a JEOL JSM-IT300 Scanning Electron Microscope to assess their morphology at 50 Pa and 20 kV. Sample morphology was evaluated with the MP-96040EXCS External Control Software. Iron presence was examined using EDS (Energy Dispersive Detection).

### 2.7. Fourier Transform Infrared Spectroscopy, FTIR

FTIR was performed on all samples to analyze their spectra where functional groups and potential intermolecular interactions can be potentially identified. The Cary 630 FTIR Spectrometer from Agilent Technologies was used, where spectra were obtained from 650 to 4000 cm$^{-1}$, using a resolution of 16 cm$^{-1}$. Four scans were used per sample, and the quartz crystal was cleaned with isopropanol between samples.

### 2.8. Thermal Analyses

Thermal stability of samples was studied by thermogravimetric analysis, TGA, and differential scanning calorimetry, DSC. These analyses were carried out to assess potential changes in the interactions between different matrix components. Temperature calibration was performed by measuring standard Curie point of nickel in open platinum crucibles, under a dry nitrogen purge flow of 20 mL/min, at a heating rate of 10 °C/min, a modulation period of 200 s, and an amplitude temperature of ± 5 °C. Analyzes were carried out in a Pelkin Elmer STA 8000 model (DT/TGA Analyzer), with 10 mg samples, calibrated

according to ASTM E968 standard, from 20 °C to 600 °C, and a heating rate of 10 °C/min, with Pyris Manager software.

### 2.9. Swelling Degree

Degree of swelling determines structure stability after each digestion, and it is attained by water absorption capacity with the interstice of pectin structures [19]. Samples were weighed before digestions, after each digestion they were dried for 5 min at 37 °C on cellulose paper and weighed again, according to methodology proposed by [24]. Degree of swelling was determined through Equation (3).

$$\% \text{ Swelling} = ((Wh - Wd)/Wd) * 100 \qquad (3)$$

where Wh and Wd are the weights of hydrated and dehydrated samples before digestion, respectively.

### 2.10. Iron Release Quantification

The amount of iron in samples after in vitro digestion was quantified to assess stability and iron retention capacity of the structures. Briefly, 250 mg of sample was diluted in tubes with 5 mL of each simulated digestive fluid (water, saliva, gastric, and intestinal). Tubes were centrifuged at 10,000 rpm for 15 min, and samples were analyzed to quantify iron concentration (ppm) in solutions, using a spectrophotometer at wavelengths of 529 and 940 nm (Thermo Scientific, model iCAP 7000). Prior to measurement, an acid digestion was performed with nitric acid ($HNO_3$), hydrochloric acid (HCl), and hydrogen peroxide ($H_2O_2$) according to standard methods (EPA 3050 and 3052). The digested solution was filtered through a 0.45 μm filter. Calibration curves were constructed from a multielement standard ICP solution 6VIII, grade Trace CERT Certipur (Merck—Millipore Sigma Aldrich, St. Louis, MO, USA), at concentration of 100 mg/L as described in our previous publication [25].

### 2.11. Statistical Analysis

For swelling and in vitro iron release analyses, values are reported as average ± standard deviation (n = 4). Analysis of Variance (ANOVA) was used to determine statistical significance, and pairwise comparison was carried out with Tukey test with a 95% confidence level ($p < 0.05$).

## 3. Results and Discussion

Pectin was used in native and oxidized forms, with the latter providing greater possibilities for interactions with glycine and iron in ferrous bisglycinate form. Characterization of these matrices is detailed and discussed below.

Solubility and stability of ferrous bisglycinate are pH dependent; it is unstable at basic pH due to the breaking of the chelate bonds, and at neutral pH, Gly-Fe chelation was not detected. Therefore, the crystals were obtained at acidic pH [26]

### 3.1. Pectin Characterization

Methoxylation degree is determined by the amount of free and esterified carboxyl groups in polygalacturonic acid units of pectin, before and after peroxide oxidation. The percentages of free carboxyl groups (Kf) in native and oxidized pectin were (22.5 ± 2.10)% and (61 ± 1.9)%, respectively. On the other hand, the percentages of esterified carboxyl groups (Ke) in native and oxidized pectin were (11.3 ± 1.58)% and (9.5 ± 1.4)%, respectively.

As esterified carboxyl content is below 50%, pectin was considered to have a low methoxylation degree. However, oxidized pectin had a higher free carboxyl percentage, possibly due to the further oxidation of carbonyl groups into new carboxyls, according to different oxidation routes pectin can undergo [27]. No significant changes were observed in terms of the esterified carboxyls because oxidation does not affect methoxyl groups [15]. Carbonyl content, on the other hand, was found at 0.4 ± 0.001 mol/g, which corresponds to a high oxidation degree [27].

### 3.1.1. Matrix Morphology

Differences in matrix morphology, and potential changes due to digestion were analyzed through scanning electron micrographs, complemented with an assessment of iron distribution through EDS, as shown in Figure 1.

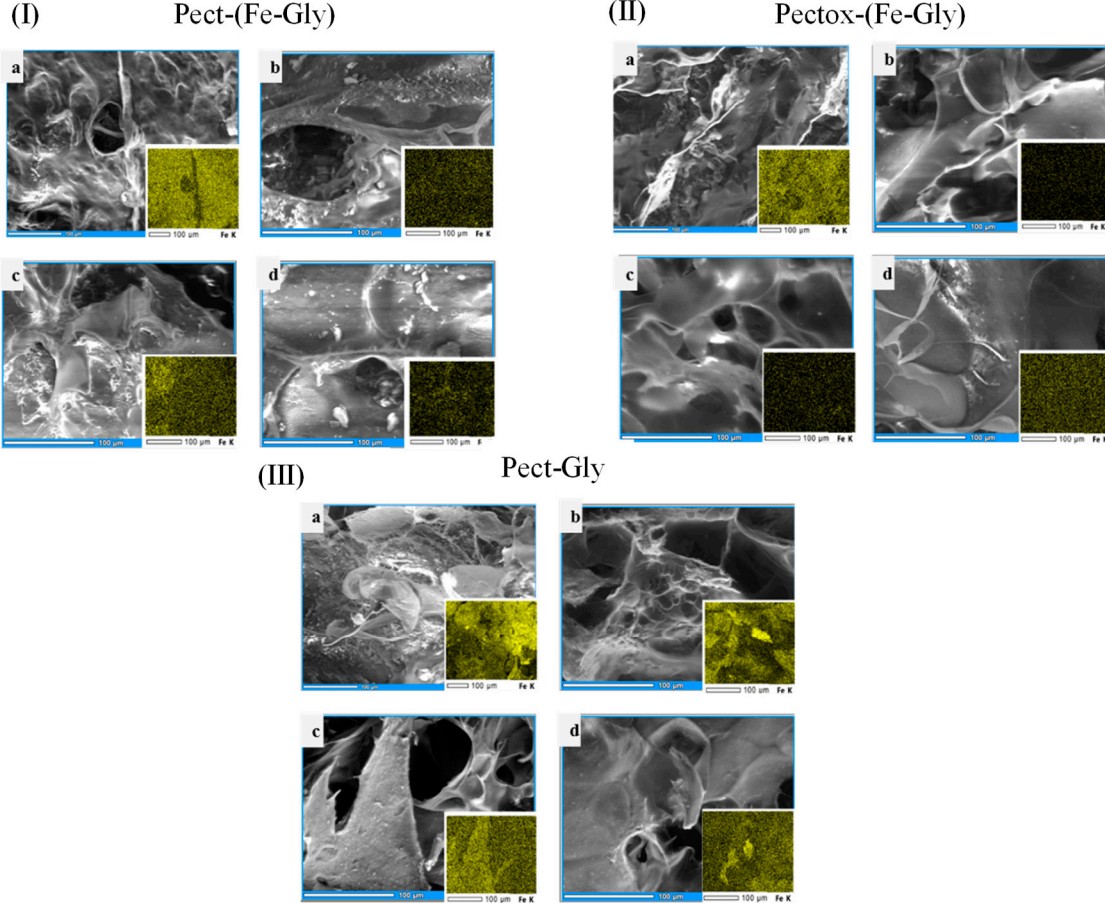

**Figure 1.** Scanning electron micrographs of different samples (**I—III**) in different simulated digestion media: (**a**) nondigested, (**b**) saliva, (**c**) gastric, and (**d**) intestinal. EDS images showing iron distribution are shown in yellow in the bottom-right insert. Calibration bar: 100 μm.

The initial morphology of the nondigested samples was porous, similar to those previously reported in the literature [28]. There were no important differences observed between the native and the oxidized pectin (Figure 1I,II). However, with digestion, there appears to be some level of erosion of the matrix, displaying some larger pores. In fact, macroscopically, after digestion, it became more difficult to handle the samples, and greater care was needed as they had the tendency to degrade. In terms of iron presence through EDS (bottom-right inserts), when comparing the chelated complex to iron only, it is clear that there was a stronger iron signal in the latter, throughout all digestion stages, while for the former, there was an important decrease in the signal after the digestion with saliva. A similar effect was observed with the oxidized pectin.

According to Ghibaudo et al., 2018, porosity of pectin has a nonhomogeneous stable structure, most likely as a result of the sample preparation process and in vitro digestion in acidic media, considering that stability of pectin gels is at pH 4 maximum. In the present work, it is implied that pectin oxidation, under the used conditions, was not significant enough to create structural changes. Furthermore, iron retention seems to be greater when using unchelated iron, possibly because iron interacts more readily with pectin through ionic links [16,28].

3.1.2. Analysis of Matrix Interactions through FTIR Spectra

The most relevant peaks that were analyzed in FTIR spectra, according to literature [27,29], were the following:

- $1000–1150$ cm$^{-1}$: C-O of pectin carboxyl and methoxyl groups.
- $1620–1690$ cm$^{-1}$: Schiff Base C=N expected to form between oxidized pectin free carbonyl and glycine amino group.
- $1640–1715$ cm$^{-1}$: C=O of pectin carboxyls.
- $1690–1750$ cm$^{-1}$: C=O of oxidized pectin free carbonyls.

In Figure 2, the signal observed between $1000–1150$ cm$^{-1}$ can be attributed to C-O stretching of pectin carboxyl and methoxyl groups. As mentioned in carboxyl quantification, native and oxidized pectin maintain similar amounts of methoxyl groups, but they can have different amounts of free carbonyls and carboxyls because, after oxidation, free carbonyls can be further oxidized to carboxyls [27,29]. Therefore, when using the unchelated versions, the peak in oxidized pectin (Figure 2b,f) has greater intensity than in native pectin (Figure 2a,e), due to carboxyls formed after oxidation, corroborated by the significant increase in carboxyl quantification previously shown. In the amino–chelated samples (Figure 2c,d), on the other hand, oxidized pectin (d) presents an attenuated signal in this range, possibly indicating combined ionic interactions with iron and charge interactions with glycine´s amino groups, and the carboxyl groups of the amino acid significantly interacting in the chelated complex.

Unfortunately, imine peaks of Schiff bases C=N ($1620–1690$ cm$^{-1}$) overlap with C=O stretching of carboxyl groups ($1640–1715$ cm$^{-1}$), so there may be the presence of both groups (C=N, C=O); however, the signal that appears in native pectin (Figure 2c,e) may be due to carboxyls that did not react with iron, so it is assumed that oxidation increases affinity for C-O-Fe interaction.

In the case of samples without glycine (Figure 2a,b), a slightly higher intensity is observed in peak at $1740$ cm$^{-1}$, corresponding to free carbonyls, a product of oxidation, and since there are no glycine amino groups, they are freely available. Furthermore, these carbonyls can be oxidized to carboxyls [27,29], as observed in a peak of oxidized pectin (Figure 2b) at $1015$ cm$^{-1}$. Therefore, both C=N and C-O-Fe Schiff base bonds appear to form, potentially with a slightly higher affinity for C-O-Fe formation, after pectin is oxidized.

On the other hand, there is a difference between amino-chelated and non-amino-chelated samples. Amino-chelates (Figure 2c,d) show greater intensity than non-amino-chelated ones (Figure 2e,f) between $1690–1750$ cm$^{-1}$ of free carbonyls; therefore, it is possible that, in chelates (Figure 2c,d), formation of Schiff base was less favored, and equal or perhaps greater amounts of C-O-Fe formation took place, than in nonchelated forms (Figure 2e,f). Moreover, in the latter (Figure 2e,f), some peaks are observed in the $1050–1150$ cm$^{-1}$ range that may correspond to carboxyl groups that perhaps did not bind to iron, or that did so to a lower extent than in the former (c,d). However, in nonchelated iron matrices (Figure 2e,f), apparently, both C=N and C-O-Fe are formed ($1620–1690$ cm$^{-1}$), as there is no previous chelation between iron and glycine, so they form their respective bonds independently. In contrast, in matrices with the chelated complex (Figure 2c,d), apparently chelated amino Gly-Fe is maintained, and influences affinity for free carboxyl or carbonyl, where it seems to have a higher affinity for carboxyl-forming C-O-Fe bond ($1000–1150$ cm$^{-1}$).

A difference in free carbonyl and carboxyl groups was observed between samples of native pectin and those of oxidized pectin, but chelated Gly-Fe complex reacts less with pectin than when Gly and Fe are added separately without chelating Gly and Fe beforehand. Thus, Fe$^{2+}$, when coordinated with Gly, may partially prevent it from reacting with pectin.

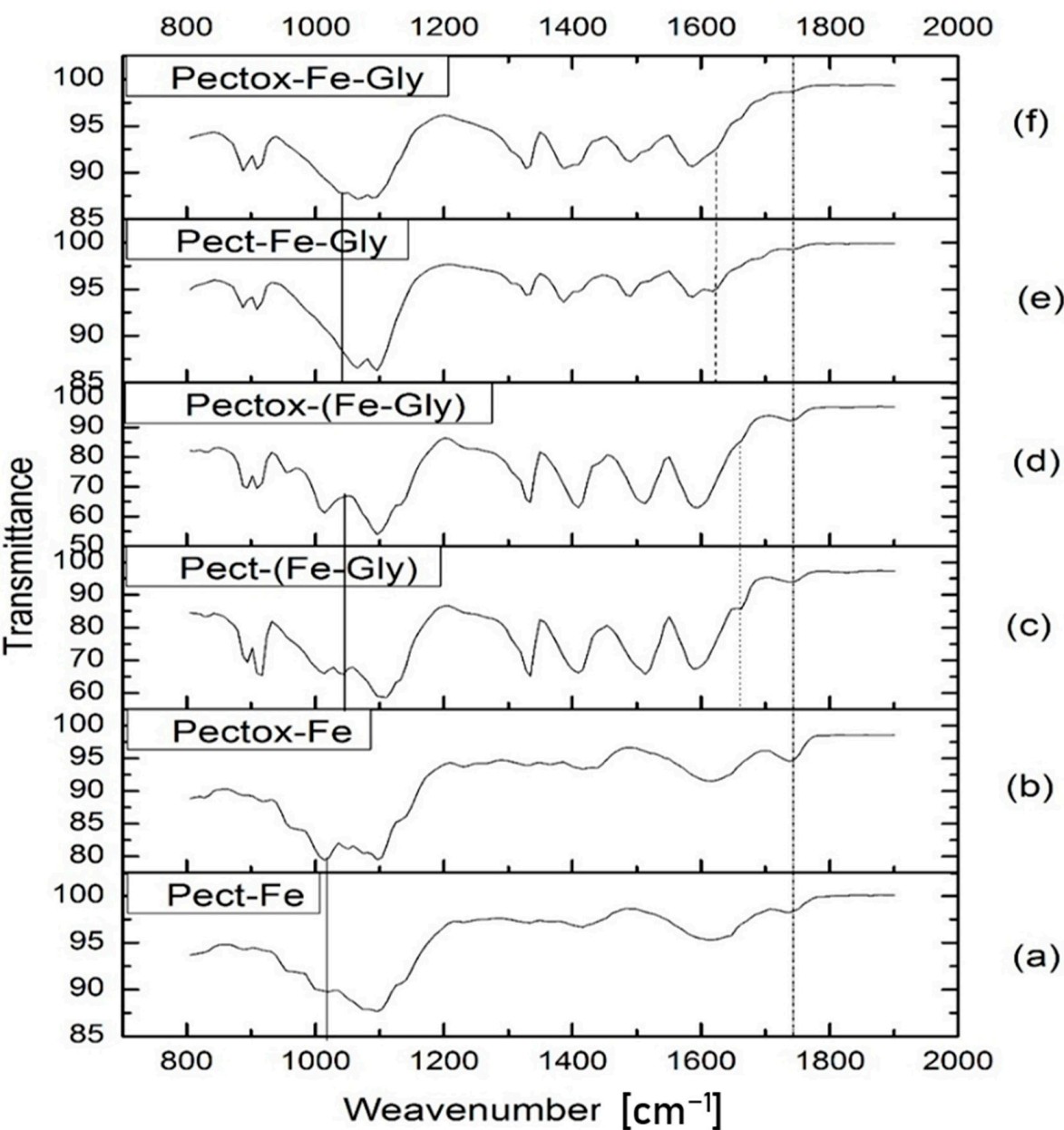

**Figure 2.** Fourier Transformed Infrared (FTIR) Spectra. (**a**) Native pectin with iron, (**b**) Oxidized pectin with iron, (**c**) Native pectin with ferrous bisglycinate, (**d**) Oxidized pectin with ferrous bisglycinate, (**e**) Pectin, with iron and unchelated glycine, (**f**) Oxidized pectin, with iron and unchelated glycine.

3.1.3. Matrix Thermal Stability Analysis with TGA and DSC

Thermogravimetric analysis, TGA, was performed to observe changes produced in thermal stability of nondigested and digested samples, which could corroborate changes in their structure and composition [30].

Figure 3a,b show four stages of degradation: a first stage from 0 to 100 °C corresponds to water evaporation; a second, from 100 to 200 °C where weight decreases slightly, relates to the beginning of pectin depolymerization [31]. A third stage, from 200 to 300 °C, a maximum degradation temperature is found, where there is a rapid decomposition due to dehydration of pectin´s hydroxyl groups; and the fourth stage, from 300 to 600 °C, represents a slow degradation due to the formation of carbon residues from pectin thermal decomposition [30].

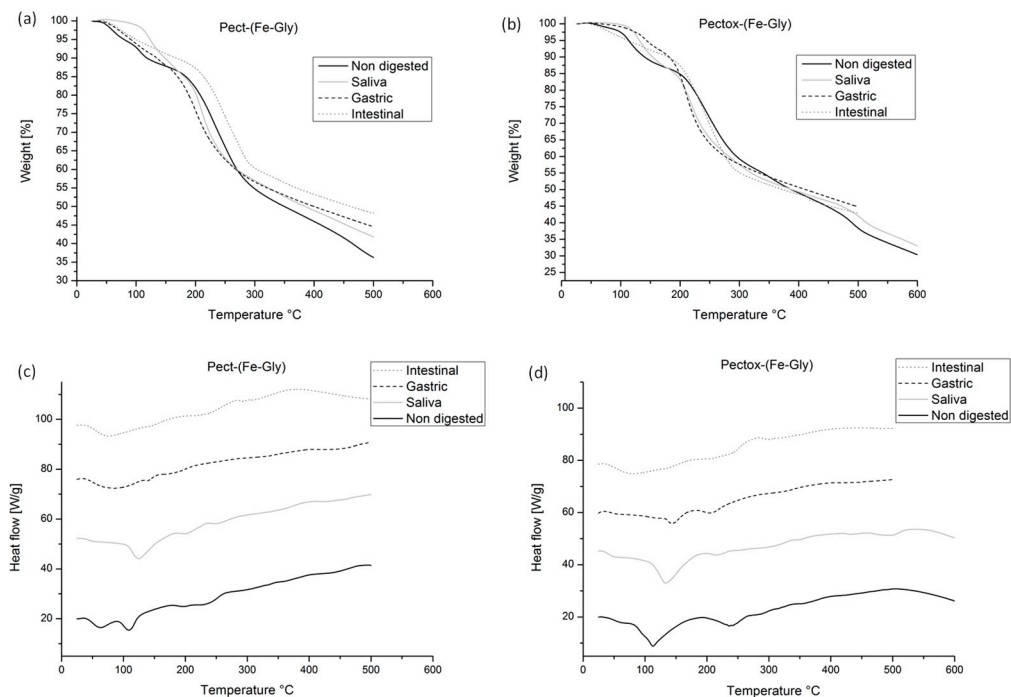

**Figure 3.** Thermogravimetric analysis, TGA (**a,b**) and differential scanning calorimetry, DSC (**c,d**) of matrices before and after in vitro digestion.

Degradation steps are similar for each fluid, except for nondigested and saliva samples, which completely degrade at 600 °C. Samples with oxidized pectin (Figure 3b) seem to have lower water content as they maintain a constant weight between 0 and 100 °C (first stage). After that, they began degradation at about 200 °C, unlike native pectin (Figure 3a), which initiates degradation at 150 °C. Thus, it is implied that samples with oxidized pectin possess greater thermal stability than their native counterparts. Furthermore, it has been previously reported that the maximum degradation temperature of pectin is around 250 °C [30], similar to what was found in this work through TGA first derivative. This temperature did not change between Pect and Pectox samples.

When comparing the undigested and digested Pect samples, there is an important difference in the percentage of residual mass at 500 °C. The undigested matrices had the lowest percentage of residual mass, followed by saliva, gastric, and intestinal digestion, respectively. This difference was greatly attenuated in the Pectox samples, even though the undigested sample had the lowest value of residual mass percentage.

DSC was also performed to assess other important thermal characteristics such as melting temperature (Tm). The higher Tm indicates a structure with an energetically favorable order, that is, more thermally and chemically stable [32–34]. DSC analysis shows two main peaks during thermal analysis of pectin; first endothermic, 100–150 °C, is attributed to evaporation of water, and a second, exothermic peak, 210–270 °C, represents thermal degradation [35]. Figure 3 shows that the peaks significantly decrease in the digested samples. In the Pect matrices, the peaks are no longer observed after gastric digestion, while, in the Pectox matrices, this happens after intestinal digestion. Thus, it can be assumed that the latter are more thermally stable, probably due to greater interactions between the pectin and the chelated complex, as implied also from TGA. The apparent more advanced degradation after digestions, particularly gastric and intestinal, may explain the fact that these had the greatest residual mass percentage at 500 °C; it is possible that, after digestion, there is a greater composition of free iron or amino-chelate, which is more thermally stable than pectin, as it has a highly crystalline structure. Ghibaudo et al., 2018 analyzed pectin–iron pearls, and found that the greatest weight loss due to thermal decomposition began above 200 °C, and reported that mass loss above 250 °C can be

explained by the interactions between an organic compound (pectin) and inorganic portions (iron) within the matrix, which would require higher calcination temperatures to remove all organic matter [35].

### 3.1.4. Swelling Degree

The swelling degree was assessed to evaluate the water absorption capacity of the different matrices, and how it changes after digestion. This analysis gives an insight into matrix stability throughout different stages of the gastrointestinal path [16]. Percentages of swelling for native and oxidized pectin in each of the simulated digestive fluids (saliva, gastric, and intestinal) are displayed in Figure 4.

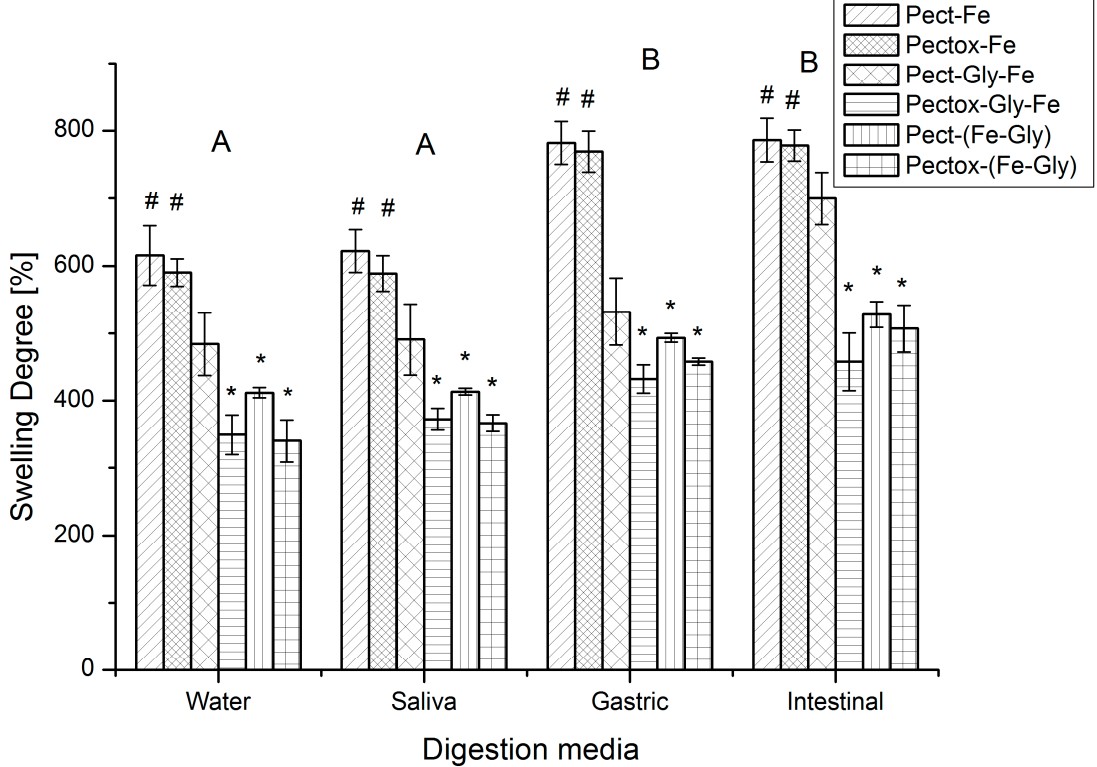

**Figure 4.** Swelling of native and oxidized pectin matrices with iron, glycine, and ferrous bisglycinate, in four different digestive fluids: water, saliva, gastric, and intestinal. (*) and (#) represent compositions that do not present statistically significant differences between them under a given digestion media. Digestive treatments with the same letters (A or B) do not present statistically significant differences ($p > 0.05$).

According to ANOVA, oxidation significantly lowered swelling degree of the matrices, with respect to those that contained native pectin ($p < 0.05$). This indicates that oxidation provides greater stability to pectin as seen in TGA and DSC analyses. Nonetheless, these differences were less preponderant in the matrices with the amino–chelated complex and became statistically equal after intestinal digestion. In the case of iron only (Pect-Fe and Pectox-Fe), there were no significant differences between native and oxidized pectin. The greater differences were found between the samples with the nonchelated glycine and iron (Pect-Gly-Fe and Pectox-Gly-Fe).

Different studies have previously explored pectin colloidal gels crosslinked with iron, and have demonstrated that this cation produces one of the most stable forms, when compared to other most commonly used gels such as copper and calcium [28,36]. In this case, it is believed, from the FTIR analysis, that oxidation produced a greater number of carboxyl groups that could potentially participate in the ionotropic gelation, but it is possible that the number of these extra groups was not sufficient to cause a significant difference in the

degree of crosslinking, and thereby affect swelling. As glycine is introduced, it may compete with iron for the carboxyl groups for charge interactions, but, when pectin is oxidized, glycine may form Schiff bases with carbonyl groups, and further provide a greater number of carboxyl groups available for ionotropic crosslinking. This would increase matrix rigidity and, in turn, translate into lower water absorption capacity.

When using ferrous bisglycinate, this chelated form has a complex structure that could have residual iron charges and amino groups on the surface of the structure that could interact with pectin in different ways: (a) ionotropic crosslinking, (b) charge interactions between aminos and carboxylic acids, and (c) Schiff bases between aminos and carbonyl groups. The first two interactions can occur with both types of pectin, while the latter would take place in oxidized pectin. The small differences in swelling between Pect-Gly-Fe and Pectox-Gly-Fe are an indication that the formations of imines via Schiff bases are secondary, perhaps being less favorable due to steric effects [37].

Regarding the effects of the digestion media, on the other hand, there were no differences between water and saliva for all formulations, possibly because pectin is not prone to enzymatic attacks by amylase. After gastric digestion, there were statistically significant increases, except for Pect-Gly-Fe, which presented a significant increase after intestinal digestion. This increase in swelling correlates to greater degradation through the weakening of the molecular interactions in the matrix, which are visualized as some erosion in SEM images (Figure 1). It is then clear that the formulations that presented greater stability through the gastrointestinal path were those with the amino–chelated complex, which correlates to their greater thermal stability.

### 3.1.5. Iron Release

Iron release in different digestion media is shown in Figure 5. In this phase, samples of unchelated iron and glycine have been omitted, since the main objective is stabilization and characterization of ferrous bisglycinate with pectin, or pectin with iron directly.

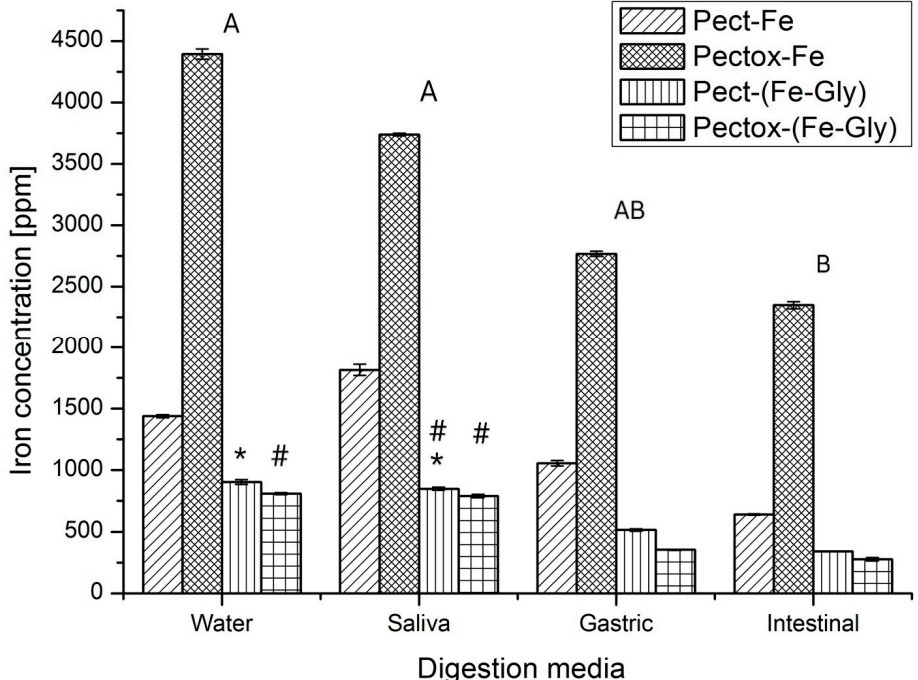

**Figure 5.** Iron Release from samples of native and oxidized pectin, with iron (Pect-Fe, Pectox-Fe) and ferrous bisglycinate (Pect-(Gly-Fe), Pectox-(Gly-Fe). (*) and (#) represent compositions that do not present statistically significant differences between them. Digestive treatments with the same letter (A or B) do not present statistically significant differences ($p > 0.05$).

Matrices with nonchelated iron displayed a rapid iron release, starting from water, particularly, those with oxidized pectin, which presented subsequent significant decrease in iron release after each digestion step. Meanwhile, Pect-Fe had greater iron retention (lower release), with maximum release during saliva digestion. This indicates that, despite having similar swelling capacity and morphology, Pect-Fe and Pectox-Fe had different behaviors, where oxidation weakens iron interactions with the matrix. When using the chelated complex, iron release decreases up to four times compared to Pectox-Fe matrices. Pect-Gly-Fe samples had a maximum release during gastric digestion, while Pectox-Gly-Fe matrices had a more gradual decrease in this parameter. In fact, these last hydrogels presented the greatest retention throughout all digestion conditions, probably due to their lower water swelling capacity resulting from stronger iron–matrix interactions. This fact may explain the lower iron EDS signals observed after digestion when it is amino-chelated as it is possible that only superficial iron is released, while the rest remains trapped in the internal structure. Importantly, Pect-Gly-Fe has a slightly stronger EDS signal when digested under gastric conditions, corresponding to the maximum observed in Figure 5. When iron is not chelated, the matrix degrades to a greater extent, compared to chelated forms, evidenced by their large swelling capacity; consequently, internal iron is released, and a consistently strong EDS signal is observed in every digestion condition.

Ghibaudo et al., 2018 assessed iron release from pectin–iron pearls, and found that gastric digestion increased pearl pores sizes, which is attributed to greater swelling, but, on the contrary, under alkaline conditions, β-removal of galacturonic acid residues from pectin occurs, resulting in chain cleavage and a rapid loss of viscosity and gelling properties, in addition to undergoing a demethylation process. Therefore, in that case, alkaline intestinal digestion of pH 8 was the main process that altered size and structure of pectin, and it is where a greater amount of iron would begin to be released from the pearls, and further undergo oxidation to form a water-soluble Fe (III)–oligopectin complex [38]

To summarize the behavior of the different matrices based on swelling and iron release, the following can be stated:

1. Amino-chelate with native pectin increases matrix stability and iron retention within the matrix, compared with pectin–Fe. The greatest release was observed in saliva.
2. Amino-chelate with oxidized pectin increases the probability of bonding with ferrous bisglycinate because the free carbonyl group presence. It displays greatest stability and iron retention, compared to the other formulations.
3. Nonchelated iron and glycine with native pectin presented lower matrix stability and iron retention capacity than the matrices with chelated iron.
4. Nonchelated iron and glycine with oxidized pectin presented the lowest stability and iron retention capacity, when compared to the chelated forms and the non-chelated with native pectin.

When doing a global analysis on swelling and iron release, along with SEM, FTIR, TGA, and DSC analyses, the results can explain the following: first, oxidation increases probability of crosslinking as there is a greater number of free carbonyl and carboxyl groups that can react, which gives the molecules greater stability. Second, pectin probably interacts with ferrous bisglycinate, only by one of the options, either with iron and pectin carboxyl, or with glycine amino and pectin carbonyl, but not by both. The leftover is maintained, but free functional groups remain, contrary to the use glycine and unchelated iron, where each one interacts and crosslinks with pectin by both options. Therefore, since there is not enough crosslinking with ferrous bisglycinate, functional groups that are not part of the crosslink attract water and allow greater swelling since they are potential sites of electrostatic interaction with water molecules [38]. Finally, the use of pectin with iron directly, without glycine, decreases its stability compared to other samples, because glycine increases crosslinking that occurs with pectin, giving it greater stability, contrary to samples that do not contain glycine.

## 4. Conclusions

Iron administration, in its Fe$^{2+}$ state, is still an important challenge in the field of fortified foods. To tackle this problem, the present study proposed to build a system of colloidal hydrogel matrices of pectin (native and oxidized) and iron, in which the combined effect of iron–glycine interactions within a chelated complex and the interactions between this complex and pectin could help retain iron in its path through the gastrointestinal track. As the metal acts as a crosslinking agent, the stability of the matrix is affected by the form of iron presentation, be it free or in an amino–chelated form (ferrous biglyscinate), and whether pectin is native or oxidized. It is implied that, when using the amino–chelated complex, oxidation increases the probability of crosslinking as there is a greater number of free carbonyl and carboxyl groups that can react, which gives the matrix greater stability. Pectin may interact with ferrous bisglycinate through different routes: iron–pectin carboxyl, or glycine amino–pectin carbonyl. Thus, potentially, the use of pectin to create a matrix that protects iron, in its ferrous bisglycinate form, would help avoid insoluble phytates formation and organoleptic changes in foods that have been fortified with this compound, and it may increase iron bioavailability. Therefore, pectin modification by oxidation may increase matrix stability, being an alternative for a possible industrial application. Nevertheless, further in vitro cell viability studies would be needed, as well as assessing long-term matrix stability and the behavior of the colloidal hydrogels in a food matrix.

**Author Contributions:** M.J. carried out most of the laboratory work and manuscript preparation under the guidance of J.F.A.-B. and F.S., who also both helped in analyzing the data. D.O. was responsible for running the TGA/DSC analyses, while D.V. and M.L. contributed with the FTIR and SEM characterization, respectively. N.C. and V.O.-H. helped with the ICP assays for iron release. All authors have read and agreed to the published version of the manuscript.

**Funding:** This work was supported by the PoliGrant program at Universidad San Francisco de Quito (HUBi 17478).

**Data Availability Statement:** The data presented in this study are available on request from the corresponding author. The data are not publicly available due to laboratory policies.

**Acknowledgments:** This work was supported by Escola de Doctorat de Tecnologia Agroalimentària i Biotecnologia, Departament d'Enginyeria Agroalimentària i Biotecnologia, Universitat Politècnica de Catalunya, Campus del Baix Llobregat, 08860, España, and by the Office of the Dean of Research at Universidad San Francisco de Quito, Campus Cumbaya, Quito 170901, Ecuador.

**Conflicts of Interest:** The authors declare that they have no known competing financial interest or personal relationships that could have appeared to influence the work reported in this work.

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
