# Peer review of "Matrices of Native and Oxidized Pectin and Ferrous Bisglycinate and Their In Vitro Behavior through Gastrointestinal Conditions"

_colloids, doi:10.3390/colloids7020035_

Round 1

Reviewer 1 Report

This manuscript reports the influence of different matrices of native and oxidized pectin on iron (Fe-Gly) bioavailability through the in vitro digestive tract. Overall, the topic of this work is worthy of investigation and shows some interesting findings. I can recommend its acceptance in the journal of Colloids and Interfaces after careful revision. There are some major comments for the improvement the manuscript quality. 

The author mentioned in the introduction section that it is believed that the colloidal structures formed by ferrous bisglycinate and oxidized pectin can be strengthened through ionotropic gelation and Schiff base reactions between the amino group of glycine and carbonyl groups in oxidized pectin. Dose it suggests that previous research may have investigated the impact of amino acids on pectin structures to improve iron release? What is the main innovation of this study? Pls explain these and supplement the innovation of this work in introduction section.

In the M&M section, the company and the country of the instruments should be supplemented.

Pls add a space between the number and the unit, e.g., 0.2 g, 5 ml.

Lines 188-189: An error occurs to the unit of °C.

Lines 250-254: The iron content detected by EDS could be quantified using software likes ImageJ.

Characterization of the texture or mechanical property of the samples before and after digestion was recommended, to enrich the results of this work.

Pls check the format of the references according to the requirements of the journal. For example, ensuring that the first letter is capital or every letter is capital. And whether the journal name is abbreviation or full name.

Overall, the technical quality of figures needs to be improved to achieve the acceptance level of journal.

Reviewer 2 Report

The study is an interesting piece of work on assessing the influence of matrix, i.e., differently modified pectin (native, oxidized, glycine chelation) on the bioavailability of iron. The research is well-designed and the findings are relevant to the field. The following comments must be addressed to improve the readability and clarity of the inferences deduced in this work:

General comments:

- Space between two words is missing in many sentences throughout the manuscript, which is to be corrected.

- Time quantities should be mentioned in numerical form. For eg., Line 122: "two hours" to be replaced by "2 h". This aspect should be checked throughout the manuscript. The SI units must be mentioned using the standard symbols (h, min instead of hours and minutes).

- The degree symbol of temperature units is underlined in all the instances throughout the manuscript, which is to be removed. 

Specific comments:

- Introduction: The authors must clearly delineate and explain the rationale and relevance of adopting each of the three matrices for the delivery of iron, i.e., amino acid chelation, ionotropic gelation, and pectin oxidation towards enhancing the bioavailability of iron. A statement of hypothesis should be more clearly mentioned in the Introduction section.

- Materials and methods:

* Section 2. Since commercial food-grade citrus pectin is used, mention the manufacturer's claim on the type (LMP or HMP), grade, and purity of pectin.

* Section 2.2. Define the notation, Kf. Line no. 121: "were" to be replaced by "was". 

* Section 2.3. Line 131: Mention whether it is %w/v, v/v, or w/w. Line 139: "were" to be replaced by "was".

* How was the ionotropic gelation performed to form the Pectin-Fe matrix? What is the stoichiometric ratio between pectin and iron (cation)? In other words, what was the concentration of ferrous sulphate used for the experiments?

* What is the physical state of the unchelated pectin-based matrices? Was it extruded in the form of beads?

* Section 2.5. Line 161" "were" to be replaced by "was". Line 170: The full forms of SEM, FTIR, TGA, and DSC should be mentioned.

* Section 2.6: Mention whether sputter coating was done and the magnification at which the images were visualized.

* Section 2.7: Mention the no. of scans performed, details of the FTIR's crystal material, and the method of cleaning the crystal between different samples.

* Section 2.8: Is the instrument a DT/TGA analyzer? If these are different, mention the make and model of both the DSC and TGA instruments.

 * Section 2.10. Line 200: "were" to be replaced by "was". Line 203: Mention the wavelength used for the spectrophotometric analysis.

Results and Discussion:

* Section 3.1: Line 234: How was the carbonyl content determined?

* Figure 1: Why were the following samples not considered for the SEM analysis, as they have been considered in most of the other analysis: Pectox-Fe-Gly, Pectox-Fe and Pect-Fe-Gly.

* Section 3.1.1. Lines 247-250: To which of the samples does the mentioned inference correspond? Please mention clearly.

* Section 3.1.3. The rationale for analyzing the thermal stability is neither well justified nor explained in the discussion section. Because the chances that the digested samples will be subjected to such high temperatures are less. Even the undigested samples may not be exposed to a temperature of more than 200-250 degree celsius in processed food products. Then, what is the reason for choosing the maximum temperature for DSC and TGA analysis as 600 degree Celsius?

* The discussion section should state the effect of each matrix on the retention, release, and bioavailability of the incorporated iron in a particular order and pattern so that it is more understandable to the readers. 

Round 2

Reviewer 1 Report

The revised manuscript has addressed most of my previous comments, so it could be acceptable for publication in journal of Colloids and Interfaces.

Reviewer 2 Report

The Authors have adequately addressed the comments raised in the first review.